# Transcriptomic and Metabolomic Insights into ABA-Related Genes in *Cerasus humilis* under Drought Stress

**DOI:** 10.3390/ijms25147635

**Published:** 2024-07-11

**Authors:** Yu Liu, Chenxue Zhao, Xuedong Tang, Lianjun Wang, Ruixue Guo

**Affiliations:** College of Horticulture, Jilin Agricultural University, Changchun 130118, China; yuliu202405@163.com (Y.L.); chen523728@163.com (C.Z.); tangxd94@126.com (X.T.); wanglianjun8892@126.com (L.W.)

**Keywords:** resistant fruit, stress, stress hormone, association analysis, regulatory genes

## Abstract

*Cerasus humilis*, a small shrub of the *Cerasus* genus within the Rosaceae family, is native to China and renowned for its highly nutritious and medicinal fruits, robust root system, and remarkable drought resistance. This study primarily employed association transcriptome and metabolome analyses to assess changes in abscisic acid (ABA) levels and identify key regulatory genes in *C. humilis* subjected to varying degrees of drought stress. Notably, we observed distinct alterations in transcription factors across different drought intensities. Specifically, our transcriptome data indicated noteworthy shifts in *GATA*, *MYB*, *MYC*, *WRKY*, *C2H2*, and *bHLH* transcription factor families. Furthermore, combined transcriptomic and metabolomic investigations demonstrated significant enrichment of metabolic pathways, such as ‘Carbon metabolism’, ‘Biosynthesis of amino acids’, ‘Biosynthesis of cofactors’, ‘Phenylpropanoid biosynthesis’, ‘Starch and sucrose metabolism’, and ‘Plant hormone signal transduction’ under moderate (Mod) or severe (Sev) drought conditions. A total of 11 candidate genes involved in ABA biosynthesis and signaling pathways were identified. The down-regulated genes included *secoisolariciresinol dehydrogenase-like* and *PYL2*. Conversely, genes including *FAD-dependent urate hydroxylase-like*, cytochrome *P450 97B2*, *carotenoid cleavage dioxygenase 4* (*CCD4*), *SnRK2.2*, *ABI 5-like protein 5*, *PP2C 51*, and *SnRK2.3*, were up-regulated under Mod or Sev drought stress. This study lays the genetic foundation for ABA biosynthesis to enhance drought tolerance and provides genetic resources for plant genetic engineering and breeding efforts.

## 1. Introduction

*Cerasus humilis*, a member of the Rosaceae family and the *Cerasus* genus, is an indigenous shrub fruit tree in China known for its nutrient-rich fruits, notably high in calcium [1]. Renowned for its resilience to drought, saline–alkaline conditions, cold temperatures, and barren environments, *C. humilis* is a wild fruit tree with exceptional adaptability. Its presence plays a crucial role in desert management, prevention of soil erosion, and ecological enhancement. Given its comprehensive stress tolerance, it serves as an ideal model for studying resistance within the *Cerasus* genus and the broader Rosaceae family [2,3,4]. Drought tolerance is a cornerstone characteristic of *C. humilis*. Following whole-genome sequencing and transcriptome analysis of *C. humilis* plants subjected to Mod drought and normal conditions, researchers identified tandem duplication amplification within the *LEA* gene family, driven by numerous *LTR*-containing genes. These genes are essential for plant development and environmental adaptation and are hypothesized to underlie the drought resistance observed in *C. humilis* [3]. Under drought conditions, drought-resistant *C. humilis* exhibits increased levels of antioxidant enzymes, particularly with upregulation expression of *cytoplasmic ascorbate peroxidase* (*cAPX*) and *dehydroascorbate reductase* (*DHAR*) [5]. Within *C. humilis*, violaxanthin can be converted to zeaxanthin by *violaxanthin de-epoxidase* (*VDE*), promoting ROS scavenging and enhancing photosynthetic efficiency. Overexpression of *VDE* results in enhanced root growth, elevated chlorophyll levels, and increased photosynthesis, respiration, and transpiration rates under drought stress [6]. The transcription factor *NAC1* positively modulates ABA-responsive genes in *C. humilis*, augmenting ABA sensitivity during root growth [7]. Notably, studies on the endogenous hormone response to drought in *C. humilis*, particularly on biosynthesis and ABA responsiveness under drought stress, are scarce [8].

Drought stress inhibits plant growth, severe drought disrupts photosynthesis and prevents it from functioning properly, and disrupts the equilibrium between the antioxidant system and reactive oxygen species (ROS) production, whereas excessive ROS levels can damage cell membrane integrity and hinder enzyme activity [9,10]. Severe drought conditions, denoted as Sev, can reduce plant yield and even lead to plant mortality, significantly affecting the fruit industry [11,12]. In 2022, the most serious hydrological and meteorological drought was recorded in the Yangtze River Basin in China, lasting 77 days at moderate (Mod) drought levels or higher. This event affected 6.0902 × 10^6^ ha crops and incurred direct economic losses amounting to CNY 51.28 billion (https://www.gov.cn/xinwen/2023-01/13/content_5736666.htm accessed on 7 May 2024). As the largest fruit producer globally, China commonly cultivates fruit trees in arid regions [13]. Hence, investigating drought stress in fruit trees is crucial to ensure secure production. Hormonal regulation is critical for enabling fruit trees to withstand drought stress, as evidenced by studies in apples [14,15], strawberries [16], grapes [17], and other species [18]. The topical application of GR24, an analog of strigolactones (SLs), during drought stress enhances chlorophyll content, induces stomatal closure to mitigate photosynthesis inhibition, suppresses malondialdehyde (MDA) and reactive oxygen species (ROS) production to alleviate oxidative damage, and regulates the expression of Ca^2+^ signaling-related genes, thus enhancing drought tolerance in apple seedlings [14]. Similarly, exogenous melatonin (MT) application in loquat seedlings under drought stress inhibited MDA accumulation, retarded chlorophyll degradation, influenced carbon metabolism pathway-related genes, and modulated genes associated with the Ca^2+^ signaling pathway. Moreover, exogenous MT affects the expression of genes related to the biosynthesis of indole-3-acetic acid (IAA), endogenous MT, gibberellin (GA3), cytokinin (CTK), and abscisic acid (ABA) [19]. In strawberry plants subjected to drought stress, treatment with methyl jasmonate minimally affects growth and morphological traits; however, it significantly alters secondary metabolism, particularly anthocyanin synthesis and accumulation [20].

ABA is a vital hormone that is crucial for various characteristics of plant growth, including leaf abscission, seed dormancy, germination, and responses to abiotic stress [21]. During drought stress, ABA synthesized in the leaves prompts stomatal closure, reducing water loss and CO_2_ assimilation. Additionally, ABA transportation to underground parts stimulates root growth and enhances water-uptake efficiency [22,23,24]. Augmenting the plant sensitivity to ABA and bolstering ABA biosynthesis are pivotal strategies for plants to combat drought stress [24,25]. Protein phosphatase type 2Cs gene *PP2C24/37* exerts a negative regulatory effect on ABA-mediated drought tolerance. Overexpression of *PP2C24/37* renders plants insensitive to ABA, and it interacts with the ABA receptor protein pyrabactin resistance 1-like (PYR/PYL) *PYL2/12* both in vivo and in vitro. Consequently, inhibition of *PP2C24/37* expression is crucial for increasing ABA sensitivity to drought stress [26]. Moreover, other genes, such as *PYL* [27], the transcription factor ABA Insensitive5 (*ABI5*) [28], and the synthesis gene for sucrose nonfermenting-1-related protein kinase 2 (*SnRK2*), play crucial roles in terms of ABA sensitivity, particularly in regulating stomatal switching [29]. The aldehyde oxidase 3 synthesis gene (*AO3*) converts abscisic aldehyde to ABA, and rice lines overexpressing *AO3* exhibit drought tolerance, characterized by elevated ABA levels in roots and increased yields [30]. *NCED* facilitates the conversion of 9-cis-neoxanthin to xanthoxin or the conversion of 9-cis-violaxanthin to xanthoxin during ABA biosynthesis, with millet plants overexpressing *NCED1* displaying heightened endogenous ABA levels and enhanced drought tolerance under stress conditions [31]. These findings indicate the importance of studying the ABA pathway in plant drought responses in *C. humilis*, a drought-tolerant fruit tree, which has significant research value with respect to elucidating its ABA response to drought stress.

This study associates transcriptomic (leveraging a reference genome [32]) and metabolomic techniques to explore the ABA-responsive drought stress network in potted *C. humilis* seedlings subjected to normal (Nor), moderate (Mod), and severe (Sev) drought conditions. The objective of this study was to identify candidate genes related to ABA biosynthesis and signaling in *C. humilis* under drought stress. This study establishes a genetic foundation for increasing ABA sensitivity or biosynthesis to enhance drought tolerance, offering valuable insights for plant genetic engineering and breeding endeavors.

## 2. Results

### 2.1. Phenotypic and Physiological Responses of C. humilis Plants under Drought Stress

The phenotypic and physiological traits of *C. humilis* were analyzed to verify the changes in response to varying levels of drought stress. *C. humilis* plants exhibited robust growth under Nor conditions, displaying superior branch length and thickness compared to those under Mod and Sev drought treatments. Under Mod drought conditions, only slight wilting of young leaves was observed in *C. humilis* plants, until the onset of Sev drought, when wilting of young leaves became evident (Figure 1). Furthermore, chlorophyll content (CC) and superoxide dismutase (SOD) activity exhibited a declining trend as drought stress intensified. CC decreased from 4.83 mg/g to 4.41 mg/g, with no significant reduction observed under Mod conditions, while SOD decreased from 243.25 U/g to 164.50 U/g. Concurrently, ELC increased from 8.43% to 13.03% with increasing drought severity. Notably, the ABA content significantly increased under Mod drought stress compared to both Nor conditions and Sev drought stress, reaching 889.26 ng/g. These findings highlight substantial physiological alterations in *C. humilis* plants under Mod and Sev drought conditions, with ABA exhibiting the most pronounced changes. To further elucidate the biosynthesis and signal transduction mechanisms of ABA under drought stress, transcriptome sequencing and metabolome assays were conducted on the young leaf tissues of *C. humilis*.

### 2.2. Transcriptome Analysis under Different Drought Levels

To investigate the drought-resistance mechanism of *C. humilis*, we constructed nine cDNA libraries for sequencing from tender leaves exposed to Nor, Mod, and Sev conditions, each with three biological replicates. After eliminating low-quality reads, 373,724,598 clean reads were obtained. The percentage of reads for *C. humilis* was in the range of 93.93% to 94.90%, with 89.39% to 90.53% being uniquely mapped, indicating the suitability of the selected reference genome (Appendix A). Additionally, the percentages of Q30 and GC content were 97.17% to 97.49% and 45.5%, respectively, demonstrating high-quality transcriptome sequencing data. Furthermore, Pearson’s correlation analysis demonstrated high reproducibility between biological replicates (Figure 2A).

The pairwise comparisons were performed among three cDNA libraries, including Nor, Mod, and Sev, for observing the differentially expressed genes (DEGs) in the samples exposed to varying drought levels. Significant genes (|log2FC| ≥ 1 and q < 0.05) were revealed in each comparison: 1686 and 1357 genes were subjected to the upregulation and downregulation for Mod vs. Nor, respectively; 1905 and 3606 genes were subjected to the upregulation and downregulation for Sev vs. Nor, respectively; and 676 and 1549 genes were subjected to the upregulation and downregulation for Sev vs. Mod, respectively (Figure 2B). These findings demonstrate that transcriptomic changes in *C. humilis* shoots are induced by drought stress.

### 2.3. The Functional Analysis of DEGs under Different Drought Levels

For evaluating the biological functions of the DEGs in *C. humilis* plants under drought stress, this study performed the Kyoto Encyclopedia of Genes and Genomes (KEGG) analysis, along with the evaluation of Gene Ontology (GO) enrichment (Appendix A). The results revealed significant enrichment of GO terms for DEGs in Mod vs. Nor, including ‘Transmembrane receptor protein tyrosine kinase signaling pathway’, ‘Plasma membrane’, ‘Protein phosphorylation’, and ‘Protein serine/threonine kinase activity’ (Figure 3A). Additionally, in Sev vs. Nor, GO terms such as ‘Cell wall organization’, ‘Hydrolase activity’, ‘Integral component of membrane’, ‘Polysaccharide binding’, and ‘Lignin biosynthetic process’ were significantly enriched (Figure 3B). Furthermore, the ‘Regulation of cellular respiration’ and ‘Oxidoreductase activity’ were significantly enriched in the GO terms of DEGs in Sev vs. Mod (Figure 3C).

The KEGG enrichment analysis indicated the significant enrichments in Mod vs. Nor, including ‘MAPK signaling pathway—plant’, ‘Starch and sucrose metabolism’, ‘Plant hormone signal transduction’, ‘Plant–pathogen interaction’, ‘Flavonoid biosynthesis’, and ‘Phenylpropanoid biosynthesis’ (Figure 3D). Additionally, in Sev vs. Nor, significant enrichments were observed in ‘Amino sugar and nucleotide sugar metabolism’ and ‘Cyanoamino acid metabolism’ (Figure 3E). Furthermore, ‘Galactose metabolism,’ ‘Ubiquinone and other terpenoid-quinone biosynthesis’, and ‘Linoleic acid metabolism’, were substantially enriched in Sev vs. Mod (Figure 3F).

The transcriptome analysis revealed that various pathways (‘Flavonoid biosynthesis’, ‘MAPK signaling pathway’, ‘Plant–pathogen interaction’, ‘Flavone and flavonol biosynthesis’, ‘Amino sugar and nucleotide sugar metabolism’, ‘Starch and sucrose metabolism’, ‘Fructose and mannose metabolism’, ‘Cyanoamino acid metabolism’, ‘Galactose metabolism’, ‘Linoleic acid metabolism’, ‘Ubiquinone, and other terpenoid-quinone biosynthesis pathways’) resulted in the adaptation of *C. humilis* plants to drought stress. In particular, ‘Plant hormone signal transduction’ (map04075) and ‘Carotenoid biosynthesis’ (map00906), including ABA biosynthesis and signaling pathways, were identified. The ‘Plant hormone signal transduction’ was notably enriched under Mod vs. Nor and Sev vs. Nor conditions, further emphasizing the significant role of ABA in drought stress response in *C. humilis*. 

### 2.4. Variations in Transcription-Factor Gene Expression under Drought Stress

To explore the regulation of drought-responsive genes in *C. humilis*, Mod vs. Nor identified 90 significant DEGs encoding TFs, with 43 up-regulated and 47 down-regulated genes (Appendix A). Similarly, Sev vs. Nor revealed 165 significant DEGs encoding TFs, with 94 down-regulated and 71 up-regulated genes (Appendix A). Furthermore, Sev vs. Mod identified 69 significant DEGs coding for TFs, comprising 45 genes exhibiting downregulation and 24 genes presenting upregulation (Appendix A). 

For the Mod vs. Nor comparison, the families of the top three transcription factors were WRKY (18%), C2H2 (15%), and MYB (13%) (Figure 4A). In the Sev vs. Nor comparison, the top three families were bHLH (15%), WRKY (13%), and C2H2 (12%) (Figure 4B). Finally, in the Sev vs. Mod comparison, the families of the top three transcription factors were bHLH (19%), MYB (19%), and C2H2 (12%) (Figure 4C).

### 2.5. The Metabolome Analysis of C. humilis under Drought Stress

To gain deeper insights into the metabolites of *C. humilis* across varying drought levels, liquid chromatography–mass spectrometry (LC-MS/MS) was used to detect the metabolites involved in the drought-tolerant process. Subsequently, the data were subjected to PCA. PCA results revealed notable clustering of samples under similar drought conditions, with distinct dispersion observed among samples experiencing different drought levels (Figure 5A).

The screening criteria for differentially expressed metabolites (DEMs) included (1) VIP > 1.0, (2) fold change > 2.0, or fold change < 0.5, and (3) *p* value < 0.05. In the positive ion mode, we detected 4185 DEMs (1876 down-regulated and 2309 up-regulated), 6338 DEMs (2646 down-regulated and 3692 up-regulated), and 3308 DEMs (2243 up-regulated, 1065 down-regulated) in Sev vs. Mod, Sev vs. Nor, and Mod vs. Nor, respectively (Appendix A and Figure 5B). In the negative ion mode, 2296 DEMs (695 down-regulated and 1601 up-regulated), 5542 DEMs (2512 down-regulated and 3030 up-regulated), and 3450 DEMs (1703 down-regulated and 1747 up-regulated) were detected in Mod vs. Nor, Sev vs. Nor, and Sev vs. Mod, respectively (Appendix A and Figure 5C).

The integration of metabolites detected in the negative- and positive-ion modes further facilitated KEGG analysis. The results revealed that Sev vs. Mod, Sev vs. Nor, and Mod vs. Nor exhibited co-enrichment in ‘Biosynthesis of phenylpropanoids’ and ‘Phenylalanine metabolism’ (Figure 5D–F). Additionally, Mod vs. Nor and Sev vs. Nor showed co-enrichment in the ‘Biosynthesis of plant secondary metabolites’ and ‘Biosynthesis of plant hormones’ (Figure 5D,E). Moreover, Mod vs. Nor demonstrated significant enrichment in ‘Glycerophospholipid metabolism’ (Figure 5D). Sev vs. Nor exhibited significant enrichment in ‘Flavonoid biosynthesis’ and ‘α-Linolenic acid metabolism’ (Figure 5E). Furthermore, Sev vs. Mod displayed significant enrichment in ‘Plant hormone signal transduction’, and ‘Starch and sucrose metabolism’ (Figure 5F). Comprehensive KEGG analysis revealed significant enrichment of ‘plant hormone signal transduction’ in both Mod and Sev conditions. 

### 2.6. Association Analysis of Transcriptome and Metabolome Data

The data from each sample were integrated and correlations between the transcriptomes and metabolomes of Nor, Mod, and Sev were analyzed. Transcriptome data with fold change (FC) ≤ 0.5 or FC ≥ 2 and a q-value < 0.05, and the metabolome data with fold change (FC) ≤ 0.5 or FC ≥ 2, a q-value < 0.05, and VIP ≥ 1 were adopted for further analyses (Appendix A). 

The metabolome and transcriptome analysis against the KEGG pathway database was conducted to elucidate the shared information of pathways and identify the predominant signaling and biochemical pathways. In the Mod vs. Nor comparison, 615 genes and 65 metabolites were identified, with 281 and 54 effective genes and metabolites distributed across 240 pathways, respectively. Enrichment in pathways included ‘Biosynthesis of amino acids’, ‘Plant hormone signal transduction’, and ‘Biosynthesis of cofactors’ (Appendix A). Moreover, for the Sev vs. Nor comparison, 998 genes and 170 metabolites were identified, with 394 and 104 effective genes and metabolites distributed across 256 pathways, respectively. The enrichment pathways encompassed ‘Carbon metabolism’, ‘Biosynthesis of amino acids, and ‘Biosynthesis of cofactors’ (Appendix A). In the Sev vs. Mod comparison, 360 genes and 125 metabolites were identified, with 203 and 85 effective genes and metabolites distributed across 179 pathways, respectively. Enrichment in pathways included ‘Biosynthesis of cofactors’, ‘Phenylpropanoid biosynthesis’, and ‘Starch and sucrose metabolism’ (Appendix A).

The correlation analyses of DEMs and DEGs were conducted to investigate the genes involved in ABA biosynthesis and hormone signaling pathways. The analysis revealed 14 significant genes for the ABA biosynthesis and signaling (Table 1 and Figure 6). Among these, seven genes were identified in the ABA signaling pathway of *Arabidopsis thaliana*, whereas six genes were linked to ABA biosynthesis in *Nicotiana tabacum* (Figure 6). The results showed that serine/threonine-protein kinase SAPK2 was not associated.

### 2.7. RNA-Seq Expression-Level Validation by qRT-PCR

For validating the specific data, the expression levels of 14 mRNAs involved in signal transduction and ABA biosynthesis were evaluated by qRT-PCR. Significant changes were observed in the gene expressions under drought conditions. Notably, the expression trend of *PP2C 24* differed from the RNA-seq results, potentially owing to detection limitations arising from low expression levels in RNA-seq. Additionally, inconsistencies were observed in the expression of *PP2C 56* compared to the RNA-seq data. Although the remaining 12 genes did not entirely align with the RNA-seq data, they exhibited similar expression trends (Figure 7).

### 2.8. The Integrated Analysis of Genes and Metabolites Related to ABA in Plants under Drought Stress

The combination of transcriptome and metabolome data revealed distinct alterations in metabolite and gene expression, addressing varying levels of drought stress and significantly affecting signal transduction pathways in plants. Combined protein interaction and qRT-PCR revealed 11 candidate genes. 

β-carotene is converted to Zeaxanthin by Cyt P450 97B2 or CrtR-β2, then cleaved to Violaxanthin by FAD-DUH-like, further oxygenated and cleaved to Xanthoxin by NCED 4, and dehydrogenated by SIDR-like or -i/-c DHM to abscisyl aldehyde, and finally to ABA. Upon sensing the increased ABA content, PYL2 binds to ABA and PP2C proteins to form a complex, thereby attenuating the inhibition of SnRK proteins by PP2C proteins, which results in the closure of stomata, as well as the activation of ABI 5-like 5 in response to the ABA-regulated. The resulting ABA pathways in *C. humilis* in response to drought stress were mapped in conjunction with ABA biosynthesis and signal transduction mechanisms (Figure 8).

In the Mod vs. Nor comparison, genes up-regulated during ABA biosynthesis included *-i/-c DHM* (fc = 2.84), *FAD-DUH-like* (fc = 3.79), and *Cyt P450 97B2* (fc = 4.32), whereas *SIRD-like* (fc = 0.47) and *CrtR-β2* (fc = 0.41) were down-regulated. In the Sev vs. Nor comparison, up-regulated genes encompassed *-i/-cDHM* (fc = 2.97), *FAD-DUH-like* (fc = 3.54), *Cyt P450 97B2* (fc = 2.62), and *NCED 4* (fc = 4.84), while *SIRD-like* (fc = 0.44) was down-regulated. In the Sev vs. Mod comparison, *NCED 4* was up-regulated (fc = 3.09) (Appendix A).

In the ABA signaling mechanism, *SnRK2.2* (fc = 2.07) and *ABI 5-like protein 5* (fc = 2.55) were up-regulated in the Mod vs. Nor comparison. For the Sev vs. Nor comparison, *PP2C 51* (fc = 5.18) and *ABI 5-like protein 5* (fc = 3.05) were up-regulated. In the Sev vs. Mod comparison, upregulation was noted for *SnRK2.3* (fc = 2.02), *PP2C 51* (fc = 3.31), and *ABI 5-like 5* (fc = 2.28), while downregulation was observed for *PYL2* (fc = 0.4) (Appendix A). 

## 3. Discussion

### 3.1. Metabolites Were Required for Drought Priming-Induced Drought Tolerance

Under drought stress, plant metabolites play a crucial role in protecting plants against environmental adversities and in maintaining cellular stability. Notably, metabolites, such as sucrose, abscisic acid, salicylic acid, glycine, asparagine, acetylcholine, linoleic acid, oleic acid, myristic acid, myristoleic oil acid, palmitic acid, erucic acid, and α-linolenic acid, were elevated to bolster plant resilience to drought stress. Additionally, significant enrichment was anticipated in terpenoid biosynthesis, phytohormone signaling, flavonoid biosynthesis, tryptophan metabolism, caffeine metabolism, sesquiterpene and triterpene biosynthesis, phytohormone signaling, and biosynthesis of phenylalanine, tyrosine, and tryptophan [33]. Hormones and secondary metabolites play pivotal roles in enhancing plant drought resistance [34]. Specifically, this study identified significant enrichment in pathways, including tyrosine and tryptophan biosynthesis, phenylalanine metabolism, plant hormone signal transduction, biosynthesis of phenylpropanoids, flavonoid biosynthesis, and ABC transporters, among other secondary metabolic pathways. Phenylpropanoid metabolism serves as a vital link between primary and specialized metabolism. The metabolites from this pathway, including flavonoids, lignin, and tannins, act as precursors, with flavonoids serving as antioxidants that mitigate drought-induced plant damage [35]. Tyrosine and tryptophan biosynthesis offers significant protection to proteins and enzymes [36]. ABC transporters, a class of ATP-binding cassette transporters, use ATP hydrolysis to facilitate the transportation of various molecules across membranes, including amino acids, sugars, nucleosides, vitamins, peptides, lipids, oligonucleotides, and polysaccharides. This diverse range of transported compounds serves as a crucial resource and foundation for various adaptive pathways in response to drought stress [37,38].

### 3.2. Key Genes Regulating the Response of C. humilis to Drought Stress

Glutathione metabolism, TF regulation, plant hormones, starch and sucrose metabolism, and MAPK signal transduction pathways were significantly enriched in plants under drought stress, consistent with the transcriptome results of this investigation [39]. Notably, *WRKY*, *MYB*, *WER*, *bHLH*, *GATA*, and *C2H2* transcription-factor families play pivotal roles in drought response [40]. For example, Dossa et al. suggested that *SiMYB75* from Sesamum indicum, when ectopically overexpressed in *Arabidopsis thaliana*, significantly enhanced root growth, increased ABA accumulation under stress conditions, elevated ABA sensitivity, and up-regulated genes associated with the ABA-dependent pathway [41]. Similarly, Lin et al. reported that VaMyb14 from *Vitis amurensis*, when overexpressed in *Arabidopsis*, enhanced drought tolerance and up-regulated genes involved in the ABA signaling pathway [42]. Moreover, the downregulation of most *MYB* transcription factors in our study may be related to the constrained anthocyanin synthesis pathway under drought stress. Additionally, Wang et al. [7] suggested that *ChNAC1*, an NAC transcription factor, was found to positively regulate ABA-responsive genes under drought stress in *C. humilis*. Zhang et al. [43] isolated *CaNAC035* from *Capsicum annuum*, which phosphorylates SnRK2.4, a key protein in the ABA signaling pathway. Transgenic chili pepper lines overexpressing *CaNAC035* significantly up-regulated the ABA biosynthesis-related genes *AAO3* and *NCED3*. Similarly, Meng et al. [44] demonstrated that *NAC56* activated *SnRK2D* expression under heat stress. The upregulation of *NAC25* and 56 under Mod conditions aligned with previous findings on the involvement of NAC in the ABA pathway. *WRKY52* was found to participate in ABA biosynthesis in tomatoes, by Jia et al. [45], whereas *DcWRKY33* in *Dianthus caryophyllus*, as revealed by Wang et al. [46], directly binds to the promoters of ABA synthesis genes (*DcNCED2* and *DcNCED5*), activating their expression. Han et al. [47] proposed that *WRKY40* in *Malus baccata* could upregulate downstream stress-related genes through an ABA-induced pathway, thereby enhancing transgenic plant survival. *C2H2* and other transcription-factor families are involved in the drought stress response of *C. humilis*. Further investigation of these factors could elucidate the mechanisms underlying drought stress in *C. humilis*. However, the specific relationship between these transcription factors and ABA biosynthesis and signaling requires further investigation.

### 3.3. Endogenous ABA Induced by Drought Priming Contributed to Increased Tolerance to Drought Stress

ABA primarily reduces water loss by regulating stomatal closure. In the ABA synthesis pathway, β-carotene undergoes hydroxylation at C3′ to form Zeaxanthin [48,49]. P450-type β-cyclohydroxylases may also participate in this process [50]. Zeaxanthin is subsequently oxidatively cleaved to Violaxanthin by zeaxanthin epoxygenase (*ZEP*) [48,51]. 9′-cis-Neoxanthin and 9-cis-Violaxanthin are further cleaved to flavins by oxygenation with NCED [52]. Xanthoxin is then converted to abscisyl aldehyde (ABAld) by xanthoxin dehydrogenase (*XanDH*), which is further metabolized to ABA by aldehyde oxidase (AO) [53]. Giorio et al. [54] demonstrated that overexpression of β-carotene hydroxylase 2 increased the conversion of β-carotene to zeaxanthin. Carotenoid cleavage dioxygenase comprises two subfamilies: *CCD* and *NCED* genes [55]. This enzyme plays a crucial role in the stress response [56]. XanDH, an enzyme localized in the cytosol, belongs to the short-chain dehydrogenase/reductase (SDR) superfamily and functions as an NAD(P)(H)-dependent oxidoreductase [53]. Similarly, isopiperitenol/carveol dehydrogenases, also members of the SDR superfamily, are associated with various plant processes such as secondary metabolism (e.g., lignan biosynthesis), stress responses, and phytosteroid biosynthesis [57]. In our study, the regulation of β-carotene hydroxylase 2 differed from that reported by Giorio et al. [54], warranting further exploration of its regulation in *C. humilis*. Although FAD-dependent urate hydroxylase remains unreported, the remaining findings are consistent with those of this study.

During ABA hormone signaling, upon increased ABA perception, PYL proteins form a complex with ABA and PP2C, alleviating the PP2C-mediated inhibition of downstream SnRK2 kinase, which activates *ABI5* in response to ABA action [58,59,60]. Conversely, in the absence or presence of low ABA levels, PP2C was up-regulated, inhibiting SnRK2 activity. Mega et al. [61] engineered wheat overexpressing the ABA receptor *PYL4*, resulting in heightened ABA sensitivity and improved water utilization. Soma et al. [62] revealed that B2-RAFs activated SnRK2 under mild drought stress, whereas B3-RAFs enhanced SnRK2 activity under severe drought conditions. These activated subclass III SnRK2s could affect the stomatal closure by phosphorylating slow anion channel-associated 1 (SLAC1). SRK2E has also been implicated in stomatal regulation [63]. Dey et al. [64] demonstrated that *SAPK9* (a subclass III SnRK2 family member) promoted osmoregulation and stomatal closure. Its overexpression reduced ROS levels and caused membrane damage. Lou et al. [65] suggested a similar function for *SAPK2*. Li et al. [66] found that *SnRK2.3* and *SnRK2.4* that were controlled by ABA could interact with *SlSUI1* to govern the ROS scavenging and accumulation during stomatal closure, protecting plant water content and enhancing photosynthesis. Liu et al. [26] identified *PP2C 24* as a negative regulator in ABA-mediated drought tolerance. These results were consistent with prior results, except for the unexpected upregulation of *PP2C 51* under severe drought stress. This contradicts the known role of the PP2C family as negative regulators of the ABA pathway, warranting further investigation to confirm the regulatory mechanism.

In addition, our study revealed the upregulation of genes associated with JA in the phytohormone signaling pathway. Numerous studies have reported a correlation between JA and ABA regulation. Further research is warranted to investigate the regulation of JA [67,68,69].

## 4. Materials and Methods

### 4.1. Plant Materials and Drought Treatments

One-year-old seedlings of *C. humilis* cv. ‘Ji’ ou No.1′, with uniform size were potted and grown in the germplasm resource nursery of *C. humilis* at Jilin Agricultural University, located in Changchun, Jilin province, China (coordinates: 125°25′33″ E, 43°49′1″ N, elevation: 207 m above sea level). The variety ‘Ji’ ou No.1′ has the characteristics of cold resistance and drought resistance, and passed the variety examination and approval in 2021. The pots, sized 15 cm × 20 cm, contained a combination of garden soil, sand, and charcoal soil in a ratio of 2:1:4. In July 2022, the plants underwent three degrees of water stress: Nor (60% ≤ soil relative humidity R), Mod (40% < R ≤ 50%), and Sev (30% < R ≤ 40%). Six replicates of each treatment were collected after receiving the drought treatment. Tender leaf blades were promptly harvested and sent to the laboratory for rapid freezing in liquid nitrogen, followed by storage at −80 °C for subsequent analyses. Each replicate weighed at least 0.3 g. Each treatment was subjected to ABA content assay (six replicates), transcriptome analysis (three replicates), metabolome analysis (six replicates), and qRT-PCR (three replicates). Functionally mature first, second, and third leaves below young leaves were collected for each treatment. Functional leaves were collected under different treatments; the leaves collected under the same treatment were mixed, and were divided into three equal parts for measuring chlorophyll content (CC), relative electrolyte conductivity (ELC), and superoxide dismutase activity (SOD). All analyses were repeated three times.

### 4.2. Measurement of Physiological Indices of Drought Stress

Chlorophyll measurements were performed based on the technique outlined by Yang et al. [70] with certain modifications. Samples weighing 0.1 g were soaked overnight in 10 mL anhydrous ethanol, diluted threefold, and analyzed using a visible spectrophotometer (722G, Shanghai Yidian) to determine the absorbance values at 649 nm and 665 nm. Chlorophyll *a* (C*a*) (C*a* = 13.95 × A_665_ − 6.88 × A_649_) and chlorophyll *b* (C*b*) (C*b* = 24.96 × A_649_ − 7.32 × A_665_) were determined. Chlorophyll content (CC) was computed by adding CC*a* and CC*b*, where CC*a* = (C*a* × 0.01 × 3)/0.1, and CC*b* = (C*b* × 0.01 × 3)/0.1. The ELC was measured according to Su et al. [71] with adjustments. Samples weighing 0.2 g were soaked in 20 mL of distilled water for four hours to measure the initial conductivity (C1). After a 30-min water bath at 100 °C and cooling to ambient temperature, the final conductivity (C2) was recorded. The ELC was determined as (C1/C2) × 100%, employing a conductivity meter (DDSL-307, Hangzhou Qiwei Instrument Co., Hangzhou, China) The SOD activity was assessed by homogenizing a sample (0.5 g) with 2 mL of 0.05 mol/L phosphate buffer (pH = 7.8), which was subjected to centrifugation at 4 °C and 12,000 rpm for 10 min, and by then performing the NBT photoreduction assay with 0.05 mL of the extract. ABA content was monitored by LC-Bio Technologies (Hangzhou) Co., Ltd. (Hangzhou, China) using multiple reaction monitoring (MRM) on a UPLC-MS/MS system. Standard chromatograms and MRM methodologies were established before UPLC-MS/MS analysis. Data collection and analysis were performed using Analyst® 1.6.3 software.

### 4.3. RNA-Seq and Function Annotation

LC-Bio Technologies (Hangzhou) Co., Ltd. (Hangzhou, China) provided the reference transcriptome sequencing. Total RNA was extracted using TRIzol reagent (Thermo Fisher, Waltham, MA, USA, 15596018) according to the manufacturer’s instructions. Subsequently, mRNA was enriched and purified using Dynabeads Oligo (dT) (Thermo Fisher, San Diego, CA, USA). The fragmentation of the mRNA was achieved using the Magnesium RNA Fragmentation Module (NEB, cat. e6150, Ipswich, MA, USA) at 94 °C for 5–7 min. Reverse transcription was performed using dUTP solution (Thermo Fisher, cat. R0133, USA), RNase H (NEB, cat.m0297, USA), E. coli DNA polymerase I (NEB, cat.m0209, USA), SuperScript™ II Reverse Transcriptase (Invitrogen, cat. 1896649, Waltham, MA, USA), AMPureXP beads, and the UDG enzyme (NEB, cat.m0280, USA). Following the PCR amplification, a cDNA library with fragments of 300 ± 50 bp was obtained. Paired-end sequencing (PE150) was conducted on an Illumina Novaseq™ 6000 (LC-Bio Technology Co., Ltd., Hangzhou, China) according to the manufacturer’s instructions. 

The reads were filtered using Cutadapt (version: cutadapt-1.9) (https://cutadapt.readthedocs.io/en/stable/) (accessed on 24 November 2022). FastQC (0.11.9) (http://www.bioinformatics.babraham.ac.uk/projects/fastqc/) (accessed on 24 November 2022) was used to verify the sequence quality, such as the Q20, Q30, and GC content of the clean data. The reads from all the samples were consistent with the *C. humilis* reference genome using the HISAT2 package (hisat2-2.2.1) (https://daehwankimlab.github.io/hisat2/) (accessed on 24 November 2022). The reference genome can be found at https://ngdc.cncb.ac.cn/gwh/Assembly/20689/show (accessed on 24 November 2022). gffcompare (gffcompare-0.9.8) and StringTie (stringtie-2.1.6) were used to generate a new transcriptome. The gffcompare software is available at (http://ccb.jhu.edu/software/stringtie/gffcompare.shtml, version: gffcompare-0.9.8), and the StringTie can be accessed at (http://ccb.jhu.edu/software/stringtie/, version: stringtie-2.1.6).

### 4.4. DEG Analysis

This analysis was conducted using the DESeq2 software (version 1.22.2) and edgeR for comparisons between both samples. Genes that met the criteria of a false discovery rate (FDR) of less than 0.05 and an absolute fold change of over 2 were categorized as DEGs. Subsequently, enrichment analyses were conducted for GO functions and KEGG pathways.

### 4.5. Metabolomic Analysis

Metabolomic analysis was performed by LC-Bio Technologies Co., Ltd. (Hangzhou, China). Thawed samples were subjected to metabolite extraction using 50% methanol buffer. Specifically, 100 mg samples were mixed with 1 mL of pre-cooled 50% methanol, vortexed for 1 min, and then incubated at ambient temperature for 10 min. The extracted mixture was maintained overnight at −20 °C. Following the 20-min centrifugation at 4000× *g*, the supernatants were placed in new 96-well plates and stored at −80 °C before LC-MS analysis. The Vanquish Flex UHPLC system (Thermo Fisher Scientific, Bremen, Germany) was used to conduct chromatographic separations with an ACQUITY UPLC T3 column (100 mm × 2.1 mm, 1.8 µm, Waters, Milford, MA, USA) stored at 35 °C. The flow rate was preserved at 0.4 mL/min, with solvent A (water with 0.1% formic acid) and solvent B (acetonitrile with 0.1% formic acid) as the mobile phases. The gradient elution was programmed as follows: 0–0.5 min, 5% B; 0.5–7 min, 5% to 100% B; 7–8 min, 100% B; 8–8.1 min, 100% to 5% B; and 8.1–10 min, 5% B. Operated in both positive- and negative-ion modes, a high-resolution tandem mass spectrometer, the Q-Exactive (Thermo Scientific), was employed for detecting the metabolites that were eluted from the column. Precursor spectra (70–1050 *m*/*z*) were acquired at 70,000 resolution, targeting an AGC of 3 × 10^6^. The maximum injection time was initialized at 100 ms, and the data acquisition followed the top three configurations in the DDA mode. To achieve an AGC target of 1 × 10^5^ with a maximum injection time of 80 ms, fragment spectra were sampled at a 17,500 resolution. After every 10 samples, a quality control sample (pool of all samples) was obtained to validate LC-MS stability throughout the acquisition. The obtained MS data underwent pretreatments, such as annotation of isotopes and adducts, second peak grouping, retention time correction, peak grouping, and peak picking, using the XCMS software (version 3.4.1). The metaX toolbox (version 1.4.16), CAMERA (version 3.4.1), and XCMS in R software (version 4.0.0) were used to convert the LC-MS raw data files to the mzXML format. By matching the exact molecular mass data (m/z) of the samples with entries in the online KEGG and HMDB databases, metabolite annotation was achieved. Subsequently, PCA and *t*-tests were conducted and differential metabolites with a VIP value ≥ 1 were identified.

### 4.6. Quantitative Real-Time Polymerase Chain-Reaction Verification

For the validation of the transcriptome data, we assessed the relative expressions of 14 genes associated with ABA biosynthesis and hormone transduction pathways via qRT-PCR under varying soil relative-humidity conditions (Nor, Mod, and Sev). The sampling method used was similar to that used for the transcriptome sampling. The details of cDNA synthesis and RNA extraction are outlined in Section 4.3. The qRT-PCR primers were designed using Primer Premier 5 software (http://www.premierbiosoft.com/primerdesign/index.html, accessed on 20 April 2021), and the list is provided in Appendix A. Relative gene expression was computed using the 2^−∆∆Ct^ method.

### 4.7. Statistical Analysis

Data processing and graphing were performed using EXCEL 2019 and GraphPad Prism 8.0 software (La Jolla, CA, USA).

## 5. Conclusions

We examined the phenotype and physiology of ‘Ji’ ou No.1′ across various drought levels and noted a notable increase in ABA levels under moderate drought stress. Subsequently, we employed transcriptomic and metabolomic approaches to elucidate significant metabolic pathways and key transcription factors in *C. humilis* under drought conditions. We identified 14 candidate genes for ABA biosynthesis and signaling pathways, which were subsequently analyzed using protein interaction networks and qRT-PCR. Finally, we identified 11 candidate genes that are involved in ABA biosynthesis and signal transduction. It is predicted that the biosynthetic process of ABA begins with the hydroxylation of β-carotene at C3′, facilitated by *Cyt P450 97B2*, to form zeaxanthin. Zeaxanthin is subsequently oxidatively cleaved to Violaxanthin with the participation of *FAD-DUH-like*. 9′-*cis*-Neoxanthin and 9-*cis*-Violaxanthin are further oxidatively cleaved to xanthoxin with the help of *NCED4*. Xanthoxin is then converted to abscisyl aldehyde (ABAld) by *SIDR-like* or *-i/-cDHM*, and ABAld is further metabolized to ABA. Upon sensing ABA, *PYL2* and *PP2C 51* form a complex with ABA, which attenuates *PP2C 51*-mediated inhibition of downstream *SnRK2.2* or *SnRK2.3* kinases, thereby activating the *ABI5-like 5* response mechanism. This provides genetic resources for plant genetic engineering and breeding efforts, laying the foundation for improving ABA sensitivity or ABA biosynthesis, to enhance drought tolerance.

## Figures and Tables

**Figure 1 ijms-25-07635-f001:**
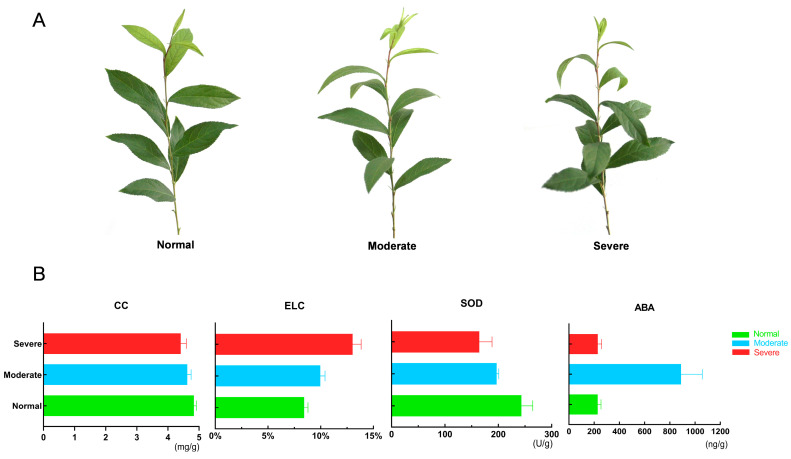
Phenotypic and physiological responses of *C. humilis* to drought stress. (**A**) is the phenotypic status of *C. humilis* plants under different drought treatments; from left to right are the states in Nor, Mod, and Sev conditions. (**B**) represents the change in chlorophyll content (CC), relative electrolyte conductivity (ELC), superoxide dismutase activity (SOD), and abscisic acid (ABA) content.

**Figure 2 ijms-25-07635-f002:**
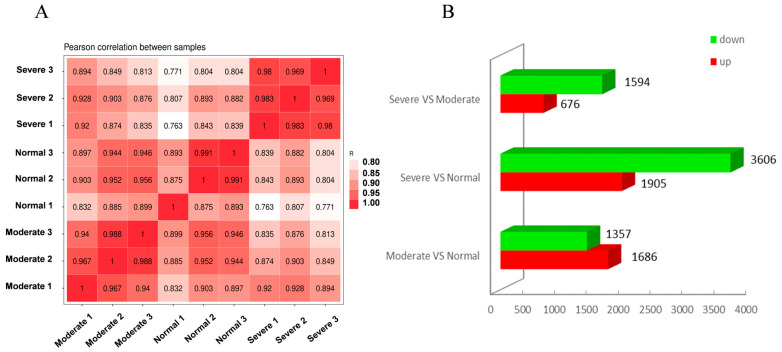
The transcriptome situations among samples under various drought-degree treatments. (**A**) is the heat map of correlation coefficients among samples under different drought-degree treatments; (**B**) is the DEGs in the different drought-degree treatment groups.

**Figure 3 ijms-25-07635-f003:**
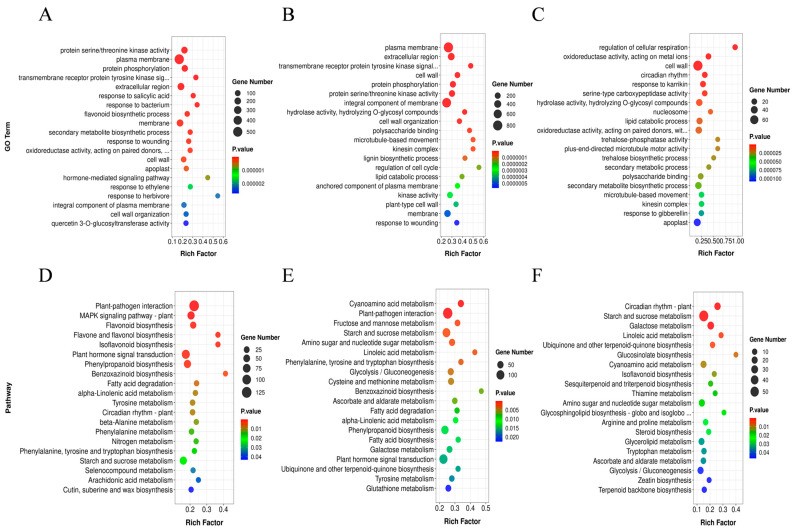
DEGs enriched in different GO terms and KEGG pathways. (**A**) represents the GO terms of DEGs in Mod vs. Nor. (**B**) is GO in terms of DEGs in Sev vs. Nor. (**C**) is GO terms of DEGs in Sev vs. Mod. (**D**–**F**) denote the KEGG pathway analysis of DEGs in Mod vs. Nor, Sev vs. Nor, and Sev vs. Mod, respectively.

**Figure 4 ijms-25-07635-f004:**
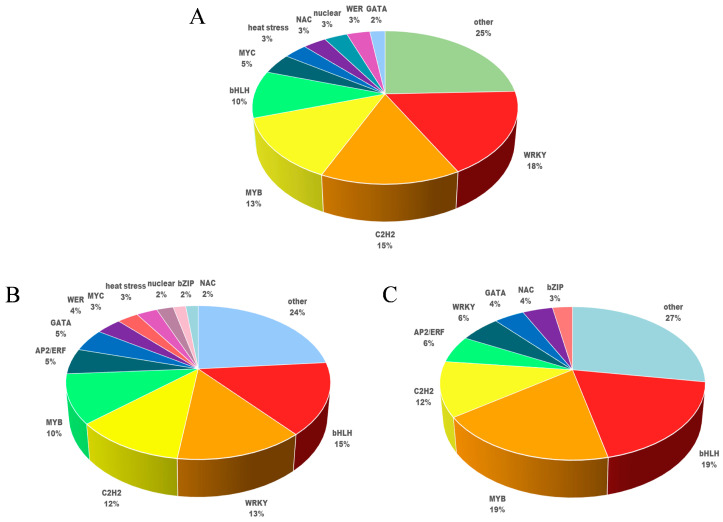
Distribution of drought-responsive transcription factor genes. (**A**) is the transcription factor classification for Mod vs. Nor. (**B**) is the transcription factor classification in Sev vs. Nor. (**C**) is the transcription factor classification in Sev vs. Mod.

**Figure 5 ijms-25-07635-f005:**
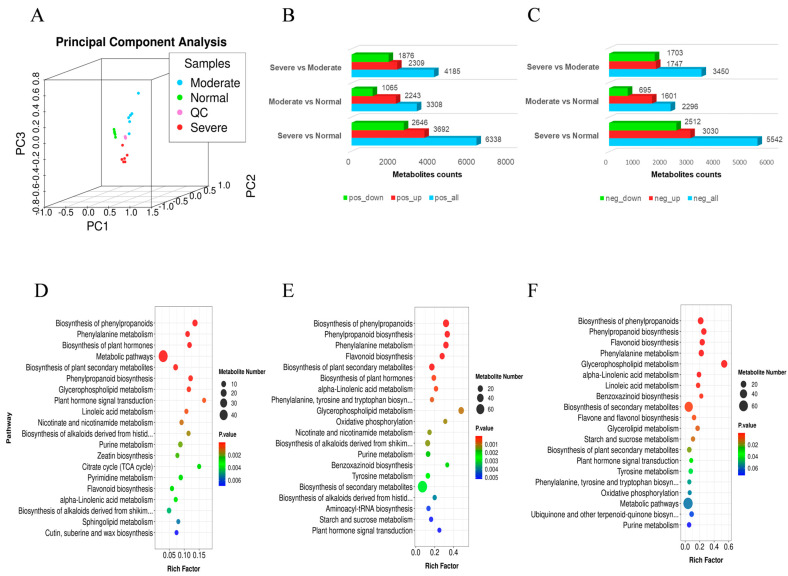
The metabolome analysis of *C. humilis* under drought stress. (**A**) is the Sample-to-sample PCA analysis. (**B**) is the DEMs between Nor, Mod, and Sev in a positive ion state. (**C**) is the DEMs between Nor, Mod, and Sev in a negative ion state. (**D**) shows the bubble plot of KEGG enrichment analysis of DEMs under Mod vs. Nor. (**E**) is the bubble plot of KEGG enrichment analysis of DEMs under Sev vs. Nor. (**F**) is a bubble plot of KEGG enrichment analysis of DEMs under Sev vs. Mod.

**Figure 6 ijms-25-07635-f006:**
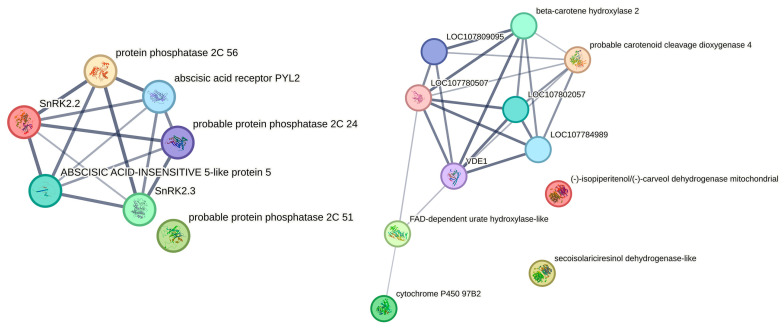
Association analysis of the genes involved in ABA signaling and biosynthesis. The left panel shows the ABA signaling association in *Arabidopsis thaliana*. The right panel shows gene metabolites associated with ABA biosynthesis.

**Figure 7 ijms-25-07635-f007:**
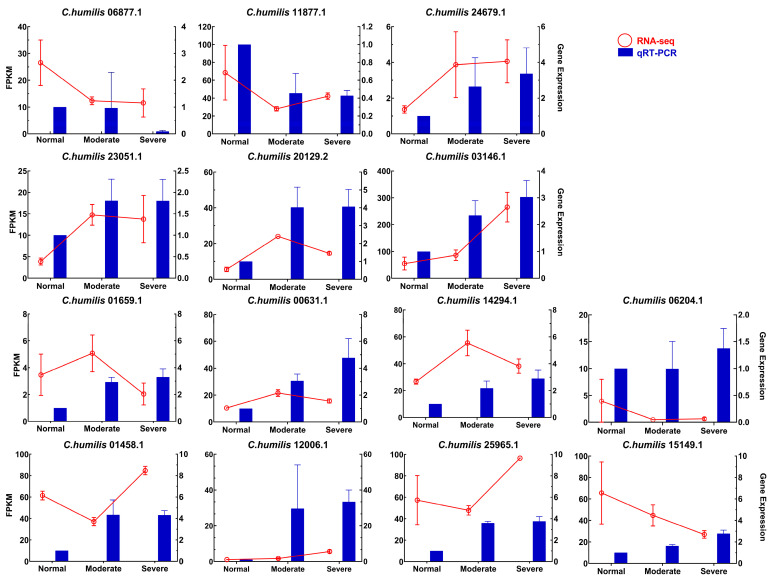
Validation of 14 candidate genes by qRT-PCR. The left *y*-axis denotes the FPKM value determined by RNA-Seq. The right *y*-axis reflects the relative gene expression levels (2^−ΔΔCt^) assessed using qRT-PCR. The *x*-axis denotes leaf samples at different drought levels. Statistical significance was determined using one-way analysis of variance (ANOVA), with different lower-case letters indicating significant differences between means (LSD, *p* < 0.05).

**Figure 8 ijms-25-07635-f008:**
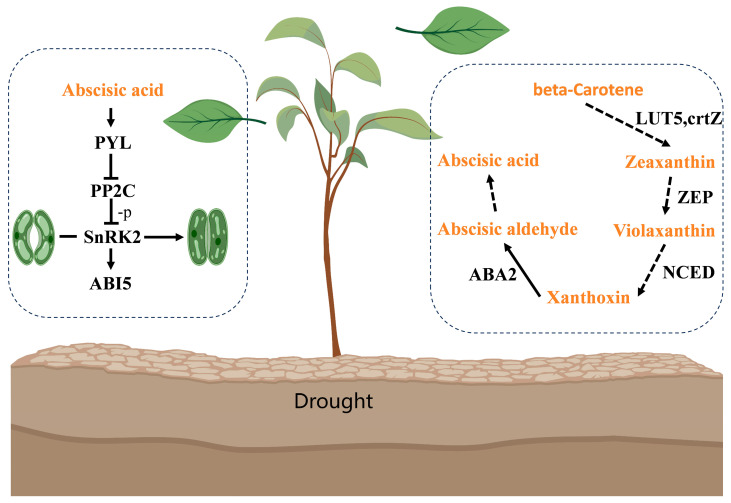
ABA biosynthesis and signaling to address drought stress. Metabolites are in orange font, and genes are in black font. *LUT5* is gene *Cyt P450 97B2*, *crtZ* is gene *CrtR-β2*, *ZEP* is gene *FAD-DUH-like*, *NCED* is *gene NCED 4*, *ABA2* is gene *SIDR-like* or *-i/-cDHM*, *PYL* is the gene *PYL2*, *PP2C* is the gene *PP2C 51*, *SnRK2* is the gene *SnRK2.2, SnRK2.3*, and *ABI5* is gene *ABI 5-like 5*.

**Table 1 ijms-25-07635-t001:** Transcript IDs, descriptions, and acronyms in the text for the 14 candidate genes.

Transcript_id	Description	Acronyms
C. humilis06877.1	Secoisolariciresinol dehydrogenase-like	SIRD-like
C. humilis11877.1	Beta-carotene hydroxylase 2, chloroplastic	CrtR-β2
C. humilis24679.1	(-)-Isopiperitenol/(-)-Carveol dehydrogenase mitochondrial	-i/-c DHM
C. humilis23051.1	FAD-dependent urate hydroxylase-like	FAD-DUH-like
C. humilis20129.2	Cytochrome P450 97B2, chloroplastic	Cyt P450 97B2
C. humilis03146.1	Probable carotenoid cleavage dioxygenase 4, chloroplastic	NCED 4
C. humilis06204.1	Probable protein phosphatase 2C 24	PP2C 24
C. humilis15149.1	Protein phosphatase 2C 56	PP2C 56
C. humilis01659.1	Abscisic acid receptor PYL2	PYL2
C. humilis00631.1	Serine/threonine-protein kinase SAPK2	SAPK2
C. humilis14294.1	SnRK2.2	SnRK2.2
C. humilis01458.1	ABSCISIC ACID-INSENSITIVE 5-like protein 5	ABI5-like 5
C. humilis12006.1	Probable protein phosphatase 2C 51	PP2C 51
C. humilis25965.1	SnRK2.3	SnRK2.3

## Data Availability

Transcriptome sequencing data are available in the SRA database of the National Center for Biotechnology Information (NCBI) under the accession number PRJNA1106200.

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
