# Peer review of "Transcriptomic and Metabolomic Insights into ABA-Related Genes in Cerasus humilis under Drought Stress"

_ijms, 2024, doi:10.3390/ijms25147635_

Round 1
Reviewer 1 Report
Comments and Suggestions for Authors
The authors used transcriptomic and metabolomic analysis to screen ABA-related genes in Cerasus humilis under drought stress. The authors performed basic analysis for transcriptomic and metabolomic data. Association analysis of transcriptomic and metabolomic data identified 14 significant genes, which were validated by qRT-PCR. The authors needs to perform English revision, detailed annotations for figures and tables, and deep excavation or elucidation for the key metabolites and key genes for ABA.
(1) Abstract: “Renowned for its fruits' significant nutritional and medicinal value, robust root system, and remarkable drought resistance”. This is not a complete sentence. Verify the grammar.
(2)Abstract: “This study primarily employed the integrated transcriptome and metabolome analyses”, to be “transcriptomic and metabolomic analyses”
(3)“1. Induction”, to “1. Introduction”
(4)“This study integrated transcriptomics, leveraging a reference genome [33], and metabolomics techniques to…”, check this sentence.
(5)The grammar needs to be modified for the whole manuscript.
(6)The icon and annotations in Figure 1A should be more clear
(7)The GO term and KEGG term should be listed with ‘’, and the english initials should be kept in upercase or lowercase to maintain consistency.
(8)“Chlorophyll a” should be “Chlorophyll a”; “chlorophyll b” should be “chlorophyll b”. a, b should be italic.
(9)In the part “2.6. Association analysis of transcriptome and metabolome data”, What are the key metabolites? What is the relationship between key metabolites with these 14 significant genes?
(10)In Fig 1B, the ABA content is highest for moderate drought treatments. Are the final key metabolites and key genes associated with this ABA result?
Comments on the Quality of English LanguageThe grammar needs to be modified for the whole manuscript.
Detailed annotations for figures and tables should be added.
Author Response
Dear Reviewer:
For all the replies in the attachment, please see the attachment.

Reviewer 2 Report
Comments and Suggestions for Authors> Abstract missing with any conclusion or final remarks from the authors. Need to add in abstract part.
> Keywords should reflect the main research part, not the words from the title or abstract.
> If title represents transcriptomic and metabolomic in the start it should start with the crop and transcriptomic or genetic part of the study not start with the drought part.
> Figure 1 in the plants we can't observe which plant are under drought stress severe or medium should indicated with A1, A2 and A3 then discussed them in the figure legends.
> Figure 2 legend is very brief need to explain more.
> Table 1 title of the table is very short need to go more in depth with it.
> Figure 7 its look with alot of information which showed very rush inside to take a look, Authors should find another way to present this figure.
Author Response

(The authors gave the same response as above.)

Reviewer 3 Report
Comments and Suggestions for Authors
This is an interesting article elucidating the genetic mechanism and factor underlying the drought stress in Cerasus humilis. The authors used the strategy of studying drought stress at various timepoints and analyzed with the approach of transcriptomic studies, metabolomic studies, RT-PCR and measurement of ABA contents. The results were good however the article can be improved with the following suggested improvements.
The authors just used one variety for evaluation of drought stress, however I suggest using multiple varieties at the same time points would be good approach to attest the results.
The authors used the genotype ‘Ji’ ou No.1’ used is tested to be drought resistance? Some background to its potential of being drought resistance will be valuable to add to the manuscript.
In Material and Methods, the authors used the term “Each treated sample was subjected to four measurements: ABA content (six replicates), transcriptome analysis (three replicates), metabolome analysis (six replicates), and qRT-PCR”. This sentence can be re-structured to convey the meaning clearly which may be appropriate to term these as approaches rather than four measurement.
In the Material and methods, the authors can describe the experimental plan more clearly by elaborating the approach mentioned as “The collected leaf mixture for each treatment was divided equally into thirds for the analysis” into thirds should be explained more clearly.
Figure 3 needs to be improved with font size to ensure clear visibility.
Figure 4. also needs to be improved for quality, it appears to be blurred.
Figure 5 consists of too much information gathered under one figure, for better understanding and visibility, the figures can be grouped into 2 Figure headings to present the data in better shape.
Table 1. Fourteen candidate genes and their transcript id. transcript_id description acronyms, the headings transcript id. transcript_id description acronyms can be capitalized for each first letter.
Figure 8, It will be valuable to explain the biosynthesis and signaling pathway involving the mentioned genes briefly.
Comments on the Quality of English LanguageOverall, the language is good, may need minor revisions for language.
Author Response

(The authors gave the same response as above.)

Round 2
Reviewer 1 Report
Comments and Suggestions for Authors
The author answered all my concerns.
Author Response
Dear Reviewer:
Thank you for your reply.